# Musculoskeletal Disorders among Agricultural Workers of Various Cultivation Activities in Upper Northeastern Thailand

**Worawan Poochada** [1] , **Sunisa Chaiklieng** [2,3,*] **and Sari Andajani** [4]

1  Doctor of Philosophy Program in Epidemiology and Biostatistics, Faculty of Public Health, Khon Kaen University, Mueang, Khon Kaen 40002, Thailand
2  Department of Environmental Health, Occupational Health and Safety, Faculty of Public Health, Khon Kaen University, Mueang, Khon Kaen 40002, Thailand
3  Research Center in Back, Neck, and Other Joint Pain and Human Performance (BNOJPH), Khon Kaen University, Khon Kaen 40002, Thailand
4  School of Public Health and Interdisciplinary Studies, Auckland University of Technology, Auckland 1142, New Zealand
*  Correspondence: csunis@kku.ac.th; Tel.: +66-93-4629696

**Abstract:** Musculoskeletal disorders (MSDs) are the most significant work-related health conditions that are experienced by agricultural workers. This cross-sectional study has investigated MSDs among agriculturalists in upper northeastern Thailand. We assessed the types of MSDs, their severity, and their frequency. There were 889 cultivating agriculturalists from four provinces who participated in this study. The majority of the participants reported experiencing mild levels of MSDs (60.48%). Predominantly, the farmers who were working on cassava, vegetable, and sugarcane plantations reported experiencing the most severe MSDs in the knees/calves (22.40%). The rice plantation workers reported the largest number of MSDs complaints. The workers on rubber plantations and in sugarcane fields were more likely to feel knee/calf pain (OR = 1.59, 95% CI = 1.05–2.39) and lower limb pain (OR = 1.97, 95% CI = 1.35–2.89) than those who were working on rice and tobacco plantations. The individuals who were working on cassava, fruit, vegetable, and corn plantations were also more likely to report knee/calf pain (OR = 1.48, 95% CI = 1.01–2.17) and lower limb pain (OR = 1.97, 95% CI = 1.37–2.84) than those who were working on rice and tobacco plantations. The MSDs that were found among those working on agricultural activities affected many parts of their bodies. The ergonomic risk needs to be assessed in order to inform plantation workers of the implications in order to improve their health and well-being and to reduce the risks of MSDs.

**Keywords:** musculoskeletal disorders (MSDs); cultivating agriculturalist; farmers





## 1. Introduction

Agricultural workers comprise the most significant proportion of people in informal employment in Thailand (58.0%), followed by those in the trade and service sector (32.2%), and those in the manufacturing sector (9.8%) [1]. Approximately 75% of these workers live and work in the northeastern part of Thailand [1]. According to the 2013 agricultural census, Thailand has nearly six million agricultural holdings, which comprise 25.9% of the total number of households in the country. Most of the agricultural holdings in the Northeast engage in crop cultivation (76.6%). In the upper Northeast of Thailand, the top four regions, where more than 97% of nationwide agricultural holdings are found, are the Khon Kaen, Roi Et, Udon Thani, and Nong Bua Lamphu provinces [2].

A systematic literature review has found that farm workers who work on agricultural plantations have the highest prevalence of musculoskeletal disorders (MSDs), at 67.8% (95% CI: 66.3–69.3) [3]. Prolonged repetitive posture [4,5], heavy cultivating, and lifting a weight of more than 10 kg [4] have been found to be risk factors for MSDs. According to Thailand's 2021 statistics [1], a prolonged repetitive posture was the most frequent cause of

MSDs (42.7%). The MSDs that were reported on agricultural plantations have also been found to affect the workers' lower backs [6,7] and lower limbs [8,9].

Previous studies on MSDs have primarily used the Nordic Musculoskeletal Questionnaire (NMQ), which asks the participants to report the bodily complaints that they have experienced in the past 7 days and the past 12 months [4,5,10–12]. The NMQ, however, does not measure the severity or the frequency of the discomfort. Chaiklieng [13] developed "an MSDs Severity and Frequency Questionnaire" (MSFQ) to be used in the industrial sector. The MSFQ has been used among rubber plantation workers in the Ubon Ratchatani province of northeastern Thailand [14]. This questionnaire yielded good reliability and was culturally relevant to Thailand's linguistic and socio-cultural contexts.

The MSFQ, which was developed by Chaiklieng [13], is a self-report survey of four questions that are used to measure the types of MSDs, the related bodily pain, and the severity and frequency of such pain. This questionnaire confirmed whether the MSDs were work-related and surveyed the workers about the MSDs that they had experienced in the past month, with reasonable validity [15,16]. The MSFQ was demonstrated with workers from a rubber plantation [14]. The highest incidence of MSDs was found among those working in crop farming [17], which was confirmed by the ICD-10 reports in the surveillance system [18]. However, Chaiklieng's MSDs assessment has never been demonstrated in any previous studies with a wide variety of cultivation types, such as those of sugarcane, cassava, rubber, corn, and tobacco, for investigation into the prevalence of MSDs. Cultivation types have different work environments that affect MSDs complaints and working posture, such as different plant sizes and different cultivation equipment. This study has examined the experiences of MSDs among those working on different types of plantations in northeastern Thailand. The findings from this study will be used to develop relevant workplace surveillance strategies and for the prevention of MSDs among plantation workers in Thailand.

## 2. Materials and Methods

### 2.1. Study Design

A cross-sectional descriptive-analytical study was designed to investigate MSDs or bodily pain experienced in the past month among workers of various agricultural types in the upper northeastern region of Thailand. This study was approved by the Khon Kaen University Ethics Committee for Human Research (registration No: HE632162).

### 2.2. Recruitment and Sampling Strategy

The participants were agricultural workers who had accessed 1 of the 33 health-promoting hospital units in the following four provinces within the upper northeastern region of Thailand: Khon Kaen province (11 hospital units), Roi Et province (7 units), Udon Thani province (10 units), and Nong Bua Lamphu province (3 units).

The sample size was calculated based on other research used in cross-sectional analytic studies on the prevalence of musculoskeletal pain in Thailand [3,19]. According to Chaiklieng et al. [3], the prevalence of MSDs among plantation workers was 0.68%. Using the 95% confidence level ($Z\alpha/2$) as 1.96, with precision not exceeding 0.04 [19], the minimum required sample size was determined to be 523 participants.

The randomized cluster sampling technique was used in the following three steps:

(1) Randomly selecting a district in each province;
(2) Then, randomly selecting sub-districts in each selected district;
(3) Finally, randomly selecting the health-promoting hospital in each sub-district.

The final number of randomly selected participants who met the inclusion criteria and gave consent to participate in this study was 889 persons. The inclusion criteria were being aged 18 years or older and having work experience in cultivating agriculture for at least one year. The exclusion criteria were having a medical history of serious injuries or congenital pathology, having a severe disability, or having a history of past surgery.

### 2.3. Structure Questionnaire

A structured questionnaire was used to collect the data. The first part of the questionnaire was based on the demographic characteristics and types of tasks performed in agricultural activity. The second part of the questionnaire was the musculoskeletal disorders (MSDs) questionnaire, which was applied to the MSFQ by using the self-reporting technique [13]. An interview was conducted with each individual participant to complete the questionnaire. The validity of the work-related MSFQ regarding pains that occurred during, or had been caused by, agricultural activities in the past month on the body areas of the neck, shoulders, upper and lower back, upper and lower arms, wrists/palms/fingers, hips/thighs, knees/calves, and ankles/feet was based on the following four primary question items [13]: (1) severity of pain, (2) frequency of pain, (3) work-related pain in the last 7 days, and (4) confirmation of the work-related pain in the past month (Figure S1). It took about 5 min to complete the questionnaire per interview.

The severity of pain, which was considered with reference to the most painful times in the past month, was classified into the following 5 levels: level 0 = no pain, level 1 = mild (annoying, interfering little with working), level 2 = moderate (pain of short duration interfering significantly with posture adaptation), level 3 = severe (persistent pain affecting the ability to work), and level 4 = very severe (persistent pain ($\geq$24 h) causing inability to perform work and affecting quality of life) [13].

The frequency of pain was considered with reference to the most frequent experience of pain, aches, or discomfort in one week, or pain that occurred several times throughout a single day. It was classified into the following 5 levels: level 0 = no pain, level 1 = 1–2 times/week, level 2 = 3–4 times/week, level 3 = once daily or every day, and level 4 = several times every day/persisting for $\geq$24 h [13].

The MSDs levels among the cultivating agriculturalists in this study were calculated by multiplying the severity level by the frequency of pain level. There were five classification levels of MSDs [13], which were as follows:

level 0 (0 points) = no MSDs;
level 1 (1–2 points) = mild MSDs;
level 2 (3–4 points) = moderate MSDs;
level 3 (5–8 points) = severe MSDs;
level 4 (9–16 points) = very severe MSDs.

The final section of the questionnaire consisted of the following four questions about how the participants were affected by any aches, pains, or discomfort: (1) Were you able to perform your daily responsibilities? (2) Did you stop working when you experienced any aches, pains, or discomfort? (3) Did you receive a Thai massage to release any aches, pains, or discomfort? (4) Did you take any medicine to relieve any aches, pains, or discomfort?

### 2.4. Data Collection

The interviews were conducted between October 2020 and March 2021 by the author [W.P.] and two research assistants. Permission to conduct the interviews was received from the relevant village chiefs where the participants resided (based on the location of the randomly selected health-promoting units). The announcements of the date and time of the study were circulated through the villages' community networks and leaders.

### 2.5. Statistical Analyses

STATA program version 14.0 was used to perform the statistical analysis. The descriptive statistics included frequency, percentage, mean, and standard deviation (SD) to describe demographic characteristics, types of tasks performed, and the severity and frequency of the MSDs experienced and self-reported by the participants. Simple logistic regression analyses were conducted to measure the association between different types of tasks specific to the specific plantation types and musculoskeletal discomfort. A $p$-value of less than 0.05 was used to determine any significant association between measured variables, the odds ratio (OR), and the 95% confidence intervals (95% CI).

## 3. Results

### 3.1. Demographic Characteristics and Types of Tasks Performed

There were 889 participants who gave consent to be subjects in this study. The majority of them were female (61.64%), with the biggest proportion aged 51–60 years old (36.45%), followed by those who were 61 years old and older (35.88%). Most of them were self-employed (91.68%). Only small number of them were the partners of workers, who had been invited to help with the tasks (2.70%). Most of the participants worked on a rice plantation (67.94%), which is a subtype of group A (rice and tobacco). A small proportion of the participants worked in corn cultivation (0.90%), which is a subtype of group C (cassava, fruit, vegetable, and corn), followed by tobacco plantations (1.35%) (in group A, rice and tobacco), and fruit plantations (3.15%) (in group C, cassava, fruit, vegetable, and corn). Further details are shown in Table 1.

**Table 1.** Number (%) of participants classified according to characteristics and grouped according to cultivation type (n = 889).

| Characteristic | Number (%) | | | | | | | |
| --- | --- | --- | --- | --- | --- | --- | --- | --- |
| | Group A | | Group B | | | Group C | | |
| | Rice (n = 604) | Tobacco (n = 12) | Rubber (n = 48) | Sugarcane (n = 80) | Cassava (n = 67) | Fruit (n = 28) | Vegetable (n = 42) | Corn (n = 8) |
| **Gender** | | | | | | | | |
| Male | 247 (40.89) | 11 (91.67) | 10 (20.83) | 22 (27.50) | 26 (38.81) | 14 (50.00) | 9 (21.43) | 2 (25.00) |
| Female | 357 (59.11) | 1 (8.33) | 38 (79.17) | 58 (72.50) | 41 (61.19) | 14 (50.00) | 33 (78.57) | 6 (75.00) |
| **Age (years)** | | | | | | | | |
| ≤40 | 24 (3.97) | 0 (0.00) | 10 (20.83) | 14 (17.50) | 4 (5.97) | 2 (7.14) | 2 (4.76) | 1 (12.50) |
| 41–50 | 109 (18.05) | 5 (41.67) | 15 (31.25) | 25 (31.25) | 22 (32.84) | 3 (10.71) | 9 (21.43) | 1 (12.50) |
| 51–60 | 228 (37.75) | 3 (25.00) | 12 (25.00) | 29 (36.25) | 25 (37.31) | 12 (42.86) | 12 (28.57) | 3 (37.50) |
| ≥61 | 243 (40.23) | 4 (33.33) | 11 (22.92) | 12 (15.00) | 16 (23.88) | 11 (39.29) | 19 (45.24) | 3 (37.50) |
| Mean (sd) | 57.08 (9.75) | 48.5 (6.00) | 49.78 (11.11) | 50.41 (10.76) | 53.58 (8.54) | 57.82 (10.21) | 56.31 (11.65) | 57 (11.62) |
| Min, Max | 21, 80 | 42, 66 | 28, 80 | 20, 79 | 36, 70 | 33, 75 | 27, 76 | 38, 73 |
| **Career** | | | | | | | | |
| Self-employed | 554 (91.72) | 12 (100.00) | 43 (89.58) | 70 (87.50) | 63 (94.03) | 27 (96.43) | 39 (92.86) | 7 (87.50) |
| Cultivation worker | 27 (4.47) | 0 (0.00) | 5 (10.42) | 10 (12.50) | 4 (5.97) | 1 (3.57) | 2 (4.76) | 1 (12.50) |
| Others | 23 (3.81) | 0 (0.00) | 0 (0.00) | 0 (0.00) | 0 (0.00) | 0 (0.00) | 1 (2.38) | 0 (0.00) |

Note: Group A comprised participants who were engaged in rice and tobacco cultivation. Group B comprised participants who were engaged in rubber and sugarcane cultivation. Group C comprised participants who were engaged in cassava, fruit, vegetable, and corn cultivation.

### 3.2. MSDs Levels

Among the 889 participants, 625 participants, or 70.30% of all of the participants, reported symptoms of MSDs in at least one part of their body. The top three most reported areas of pain were the knees/calves (39.68%), the lower back (35.68%), and the shoulders (30.08%). Of the 625 participants, 13.92% reported a severe level of discomfort, while others reported a moderate level (23.04%), and a mild level (60.48%) of discomfort. Regarding the frequency of pain, the highest proportion of participants experienced lower back pain three to four times/week (14.72%), followed by those with knee/calf pain, which was experienced one to two times/week (14.56%), and once daily or every day (13.60%) (Figure 1).

Across the levels of discomfort that were reported, the most reported location of discomfort was in the knees/calves, followed by the lower back (21.44%), and the shoulders (19.68%) (Figure 2). The 625 participants who reported MSDs confirmed that they had experienced work-related pain in the past month. Of the respondents, 83.10% had no opportunity to stop their work due to pain. The majority of them (60.86%) decided to take a drug instead of getting a Thai massage to reduce their pain (27.73%).

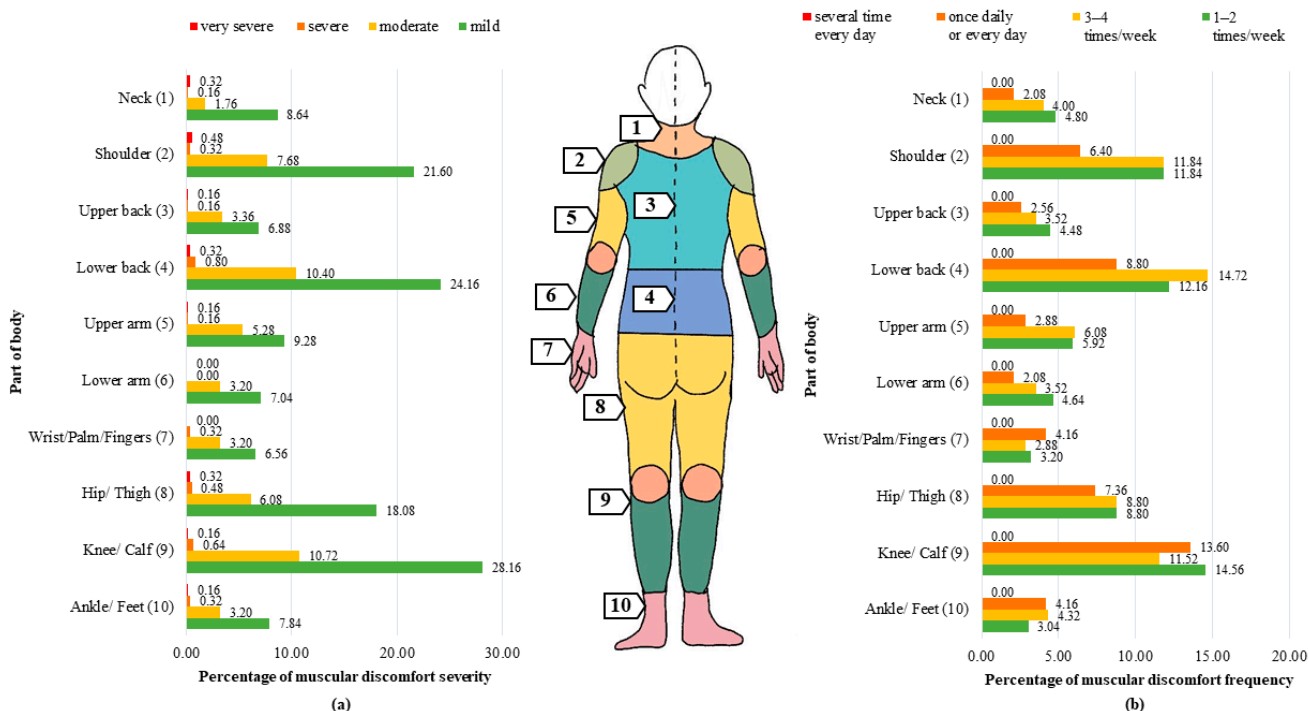

**Figure 1.** The percentages of (**a**) severity and (**b**) frequency of muscular discomfort according to parts of the body.

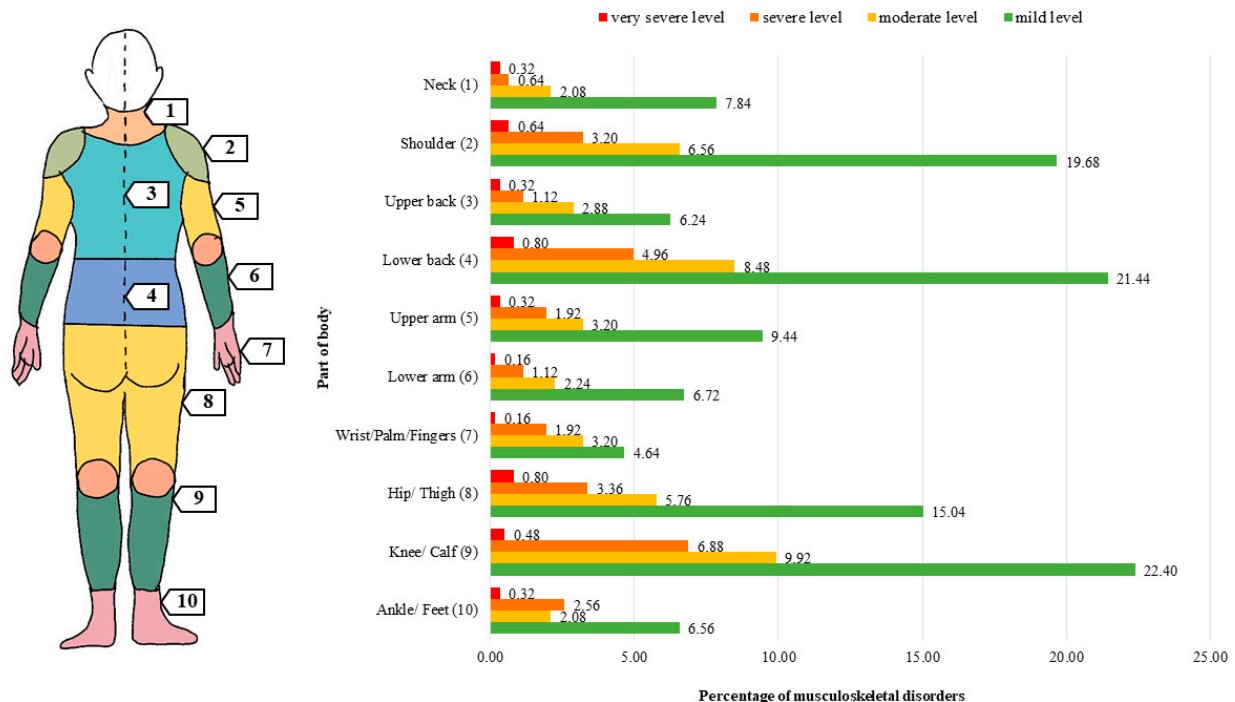

**Figure 2.** MSDs levels classified according to affected body parts.

According to the results that are based on the groups and the cultivating types, the workers from group C (cassava, fruit, vegetable, and corn farming) reported the most experience with severe to very severe levels of discomfort, followed by group B (rubber and sugarcane farming). Further details are shown in Figure 3.

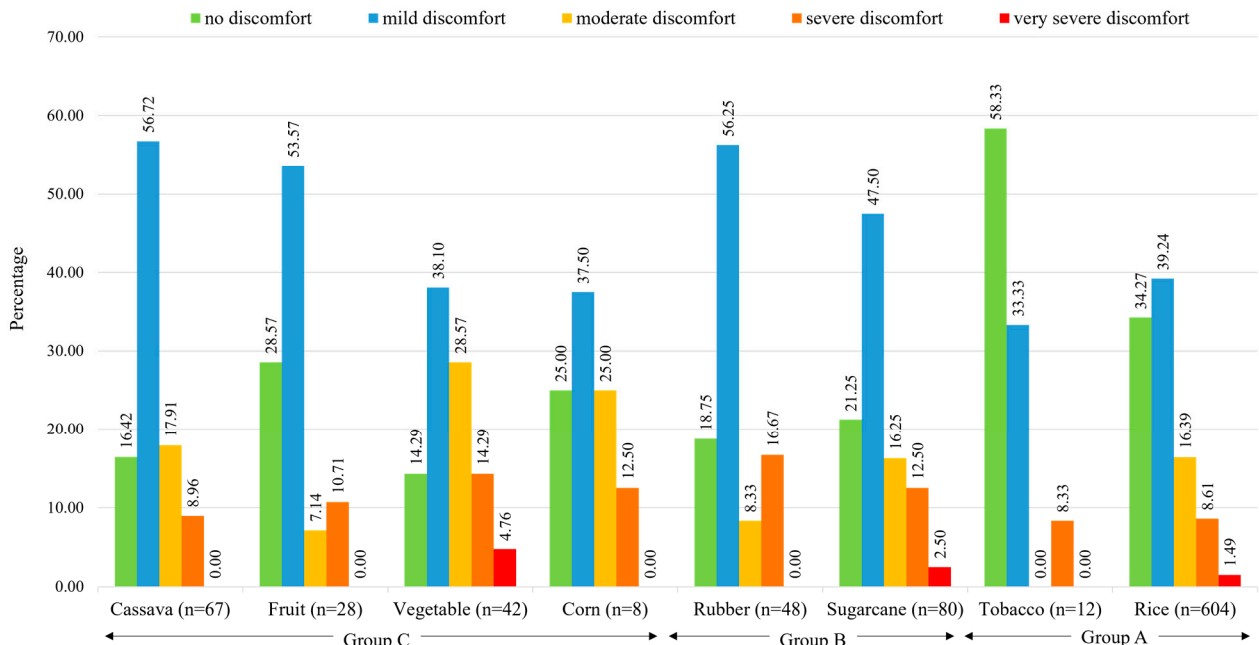

**Figure 3.** Musculoskeletal discomfort levels classified according to group and cultivating type.

According to Figure 4, most of the participants reported a mild discomfort level. A moderate level of discomfort was reported by those who were working on group C crops (vegetable (47.62%) and corn (37.50%) plants) and group B crops (sugarcane (31.25%) plants). The results showed that the levels of MSDs varied according to the tasks performed on the rice plantations. Those who were working on sowing the seeds of rice were most likely to report discomfort (90.85%), followed by the field ploughing workers (83.33%), and the mowing workers (78.13%).

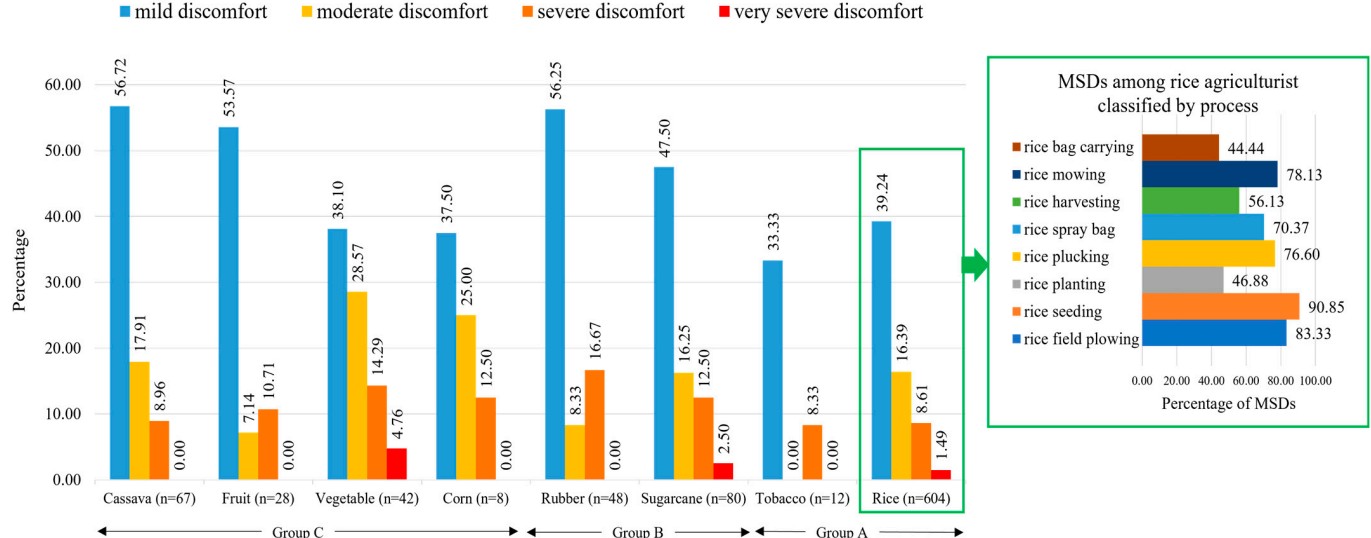

**Figure 4.** Musculoskeletal discomfort levels classified according to group and cultivating type.

The simple logistic regression analysis showed an association between the different types of cultivation and the discomfort of the knees/calves among workers in group B, which consisted of the rubber and sugarcane cultivation (OR = 1.59, 95% CI = 1.05–2.39), and those working in group C, which consisted of the cassava, fruit, vegetable, and corn cultivation (OR = 1.48, 95% CI = 1.01–2.17). The workers were likely to report discomfort in the lower limbs in group B, which consisted of the rubber and sugarcane cultivation

(OR = 1.97, 95% CI = 1.35–2.89) and group C, which consisted of the cassava, fruit, vegetable, and corn cultivation (OR = 1.97, 95% CI = 1.37–2.84). Further details are shown in Table 2.

**Table 2.** The association between cultivation type and muscular discomfort shown by simple logistic regression (n = 889).

| Area of Discomfort according to Cultivation Type | n | Muscular Discomfort | | OR | 95% CI | *p*-Value |
|---|---|---|---|---|---|---|
| | | Yes (%) | No (%) | | | |
| **Shoulders** | | | | | | |
| Group A | 616 | 123 (19.97) | 493 (80.03) | 1.00 | | |
| Group B | 128 | 26 (20.31) | 102 (79.69) | 1.02 | 0.64–1.64 | 0.929 |
| Group C | 145 | 39 (26.90) | 106 (73.10) | 1.47 | 0.97–2.24 | 0.068 |
| **Lower back** | | | | | | |
| Group A | 616 | 152 (24.68) | 464 (75.32) | 1.00 | | |
| Group B | 128 | 33 (25.78) | 95 (74.22) | 1.06 | 0.68–1.64 | 0.792 |
| Group C | 145 | 38 (26.21) | 107 (73.79) | 1.08 | 0.72–1.64 | 0.702 |
| **Hip/thigh** | | | | | | |
| Group A | 616 | 102 (16.56) | 514 (83.44) | 1.00 | | |
| Group B | 128 | 23 (17.97) | 105 (82.03) | 1.10 | 0.67–1.82 | 0.698 |
| Group C | 145 | 31 (21.38) | 114 (78.62) | 1.37 | 0.87–2.15 | 0.170 |
| **Knee/calf** | | | | | | |
| Group A | 616 | 146 (24.19) | 467 (75.81) | 1.00 | | |
| Group B | 128 | 43 (33.59) | 85 (66.41) | 1.59 | 1.05–2.39 | 0.028 * |
| Group C | 145 | 56 (38.62) | 89 (61.38) | 1.97 | 1.35–2.89 | <0.001 * |
| **Lower limbs** | | | | | | |
| Group A | 616 | 217 (35.23) | 399 (64.77) | 1.00 | | |
| Group B | 128 | 57 (44.53) | 71 (55.47) | 1.48 | 1.01–2.17 | 0.048 * |
| Group C | 145 | 75 (51.72) | 70 (48.28) | 1.97 | 1.37–2.84 | <0.001 * |

Note: Group A comprised participants who were engaged in rice and tobacco cultivation. Group B comprised participants who were engaged in rubber and sugarcane cultivation. Group C comprised participants who were engaged in cassava, fruit, vegetable, and corn cultivation. * Statistically significant at *p*-value of <0.05.

Overall, the workers in the rice fields reported experiencing a higher level of MSDs in the lower limbs compared to the other plantation workers. An exception was those who carried the rice bags, who reported more discomfort around their shoulders (Table 3).

**Table 3.** MSDs experienced by rice cultivating agriculturalists according to rice processing activity (n = 604).

| Rice Processing Activity | n | Number Experiencing Muscular Discomfort (%) | | | |
|---|---|---|---|---|---|
| | | Shoulders | Hips/Thighs | Knees/Calves | Lower Limbs |
| Rice field ploughing | 12 | 2 (16.67) | 3 (25.00) [3] | 5 (41.67) [2] | 7 (58.33) [1] |
| Rice seeding | 153 | 45 (29.41) [3] | 29 (18.95) | 51 (33.33) [2] | 75 (49.02) [1] |
| Rice planting | 160 | 14 (8.75) | 19 (11.88) [3] | 31 (19.38) [2] | 41 (25.62) [1] |
| Rice plucking | 47 | 9 (19.15) [3] | 8 (17.02) | 14 (29.79) [2] | 19 (40.43) [1] |
| Rice bag spraying | 27 | 6 (22.22) [3] | 4 (14.81) | 9 (33.33) [2] | 11 (40.74) [1] |
| Rice harvesting | 155 | 32 (20.65) [2] | 31 (20.00) [3] | 28 (18.06) | 49 (31.61) [1] |
| Rice mowing | 32 | 8 (25.00) [2] | 6 (18.75) [3] | 8 (25.00) [2] | 10 (31.25) [1] |
| Rice bag carrying | 18 | 5 (27.78) [1] | 1 (5.56) | 2 (11.11) [3] | 3 (16.67) [2] |

Note: [1,2,3] denote the top three pain areas associated with each rice processing activity.

## 4. Discussion

About 70% of the participants reported MSDs in at least one part of their body. Knee/calf, lower back, and shoulder pain were the most frequently reported. The top three areas of the body where MSDs were experienced, with regard to their severity and pain level, were also the knees/calves, the lower back, and the shoulders. This finding concurs with previous Thailand-based studies [5–7,20].

Knee/calf pain was reported by nearly one-third of the workers in each plantation group. The knee/calf area is one of the most affected areas and is an area where workers frequently experience some level of discomfort. A study by Peungsuwan et al. [4] found that 54.04% of elderly plantation workers experienced knee pain. In our study, 36.45% of the participants were 51–60 years old, and 35.88% were 60 years old and older. Our

findings concur with those of Neubert et al. [9], who found that the age of the plantation workers, namely, being older adults, was associated with lower limb pain.

This study found significant associations between the different types of tasks performed in each plantation type and the severity and frequency of the MSDs that were experienced. Those who were working on cassava, fruit, vegetable, and corn plantations (group C) had a significantly higher risk of knee/calf pain (OR 1.97, 95% CI = 1.35–2.89) and lower limb pain (OR = 1.97, 95% CI = 1.37–2.84) than those who were working on rice and tobacco plantations (group A). Moderate and higher levels of severity of discomfort were reported among those who grew vegetables (47.62%); the highest frequency of discomfort was found among those in group C who were cultivating corn (37.50%). This may relate to the cultivation of the types of crops in group C involving a high-frequency planting–harvesting cycle of 90–120 days. Moreover, working on group C plantations may require repetitive switching between sitting and standing positions, which causes discomfort around the knees/calves and in the lower limbs. A previous study showed that the tasks that required prolonged sitting, standing, and walking led to knee pain (OR = 2.39, 95% CI = 1.06–5.39) [4]. The squatting position is also associated with knee pain [21].

The workers who engaged in work on the rubber and sugarcane plantations (group B) were more likely to experience discomfort in their knees/calves (OR = 1.59, 95% CI = 1.05–2.39) and in their lower limbs (OR = 1.48, 95% CI = 1.00–2.17) than those who were working on the rice and tobacco plantations (group A). Although sugarcane cultivation has a longer planting cycle of 270–360 days, those who worked on sugarcane plantations reported experiencing the highest severity across the top two body regions. The tasks involved in sugarcane cultivation include manually cutting the sugarcane at ground level, removing the leaves, and trimming off the last mature joint. Then, the sugarcane is stacked into large piles and is loaded onto the trucks. All of these tasks are manually performed using agricultural tools. This finding is supported by Kaewdok et al. [5], who found that using agricultural tools (adjusted OR = 4.40, 95% CI = 1.18–13.79) and lifting >10 kg (adjusted OR = 2.87, 95% CI= 1.22–6.82) were significantly associated with a high prevalence of discomfort around the lower limbs (65.4%) among older farmers.

According to Table 2, the agriculturalists who worked on group A plantations (rice and tobacco cultivation) comprised the lowest proportion of agriculturalists who were experiencing muscular discomfort at moderate and higher levels. In Thailand, rice fields are used for tobacco cultivation when it is not the season of rice cultivation. Tobacco cultivation also has a similar process to rice cultivation along with a similar cycle time for growing. The majority of the participants in this study (67.94%) worked on rice plantations. They performed various activities in the rice fields, and those who sowed the rice seeds reported the highest level of MSDs in unspecified parts of the body. With regard to the body regions of discomfort (Table 3), the rice seeding activity resulted in the highest proportion of agriculturalists with discomfort in the lower limbs (49.02%). Thai rice agriculturalists perform rice seeding by carrying a rice bucket (weighing more than 10 kg) and hand sowing the rice while walking for a prolonged period of time in shallow water in muddy fields. This finding is supported by Neubert et al. [9], who found that the force of exertion in the rice seeding activity was a significant factor in knee pain. Moreover, holding or carrying a weight of more than 10 kg might overload the knee muscles and tendons [22]. This study found that differences in the cultivation type were not associated with shoulder or lower back pain. About 83.10% of workers reported having no opportunity to stop their work due to pain. The pressure of the planting and harvesting cycle to produce quality products makes it impossible for the workers to stop, rest, and recuperate. Failure to produce good crops will have a significant economic impact on the workers, their families, and the farm owners. This continuation of strenuous work while experiencing some level of pain increases the risk of a more frequent and higher level of discomfort. For example, 14.72% of the participants reported experiencing some level of discomfort three to four times/week. Lower back pain complaints were prominent among the rice farmers, especially during the

planting stage [23], as were complaints of pain in the lower back, the hip joints, and the knee joints [20].

This study found that shoulder pain was among the top three types of musculoskeletal pain (13.84%) reported by the workers from the group C plantations (cassava, vegetable, and fruit). This shoulder pain may relate to carrying; among various activities of rice planting, muscular discomfort was found to be the highest in the shoulders, due to rice bag carrying (27.78%) (Table 3). A study by Fulmer et al. [24] confirmed that, during pesticide spraying, the downward pressure from the bucket's strap was a major ergonomic problem that was causing the shoulder pain. Manually picking and reaching for fruit with the elbows positioned above shoulder height was also a major ergonomic factor that was causing shoulder pain among those who were harvesting apples [24]. The harvesting stage of planting had a high ergonomic risk of musculoskeletal disorders, which were assessed by using the ergonomics risk assessment technique [25,26].

The strong points of this study were the wide variety of cultivation types that were studied and the grouping of the cultivation type according to the cycle times of planting (field preparation for the harvesting process) and the main posture types in each harvesting process (sitting or standing). This study had the following limitations: the age- and gender-match differences of the participants, the work experience in agriculture, and the physical activity of the participants were not evaluated. The sample size calculation and the sampling were not distributed according to the cultivation type. A selection bias may have occurred as we enrolled all of the participants who engaged in any type of cultivation. There was no equal distribution of the number of participants from each plantation type. However, even though the areas of interest (four provinces) had differences in plantation types, it was likely that the main plantations were for rice cultivation in each area. In future, researchers may consider applying an equal distribution of participants in each type of plantation by focusing on specific plantation types and examining each specific task that is involved in each plantation type in more depth.

All of the MSD data in this study were based on self-reported MSDs, in which there were no physical exams conducted. Recall bias may have occurred due to the age of the participants; this study found that the highest percentage of agriculturalists were over 50 years old, or near-elderly persons. Thai agriculturalists are usually elderly people because they work in the manufacturing industry from being teenagers and come back to their hometowns when they are getting older. In order to reduce the recall bias, future studies should be designed as cohort studies, while case follow-up and investigation of the factors that are related to MSDs in the cultivating group that experiences kinds of pain, which are analyzed by multiple logistic regression, should be provided.

## 5. Conclusions

The majority of workers in various types of plantations have reported a mild level of MSDs (60.48%), followed by those experiencing a moderate level (23.04%), and a severe level (13.92%) of discomfort. The top three body parts with the highest levels of MSDs were the knees/calves (22.40%), the lower back (21.44%), and the shoulders (19.68%), among workers on cassava, vegetable, and sugarcane plantations. The highest frequency of complaints was found in rice planting. The plantation and cultivation types were associated with the MSDs that were experienced. The agriculturalists in group A (rubber and sugarcane) and group B (cassava, fruit, vegetable, and corn) were more likely to report knee/calf pain and lower limb pain. A regular and targeted ergonomics risk assessment and MSD surveillance are, therefore, crucial in preventing MSDs and for promoting the well-being and longevity of the farming workers in Thailand, especially as the workers are ageing, with the majority being between 50 and 60 years of age.

**Supplementary Materials:** The following supporting information can be downloaded at: https://www.mdpi.com/article/10.3390/safety8030061/s1, Figure S1: The MSDs Severity and Frequency Questionnaire (MSFQ) [13].

**Author Contributions:** Conceptualization, S.C.; methodology, S.C.; software, S.C. and W.P.; validation, S.C. and S.A.; formal analysis, W.P.; investigation, W.P., S.C. and S.A.; resources, W.P.; data curation, W.P.; writing—original draft preparation, W.P. and S.C.; writing—review and editing, W.P., S.C. and S.A.; funding acquisition, S.C.; project administration, S.C. All authors have read and agreed to the published version of the manuscript.

**Funding:** This research was financially supported by the National Research Council of Thailand (NRCT 6200101).

**Institutional Review Board Statement:** This study was approved by the Khon Kaen University Ethics Committee for Human Research (Registration No. HE632162).

**Informed Consent Statement:** All of the participants who engaged in the study provided informed consent.

**Data Availability Statement:** The data described in this study are accessible from the corresponding author upon request. Due to confidentiality concerns, the data are not publicly available.

**Acknowledgments:** The authors thank all of the agricultural workers for their kind co-operation throughout this study. The authors are grateful for the NRCT grant (No. 6200101).

**Conflicts of Interest:** The authors declare no conflict of interest.

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
