# Peer review of "Musculoskeletal Disorders among Agricultural Workers of Various Cultivation Activities in Upper Northeastern Thailand"

_safety, 2022_

Round 1

Reviewer 1 Report

This is an excellent paper with important findings. Please see below for some suggested improvements.

Line 43 - Typo, should be prolonged

Line 53 - Missing word at the beginning of the sentence?

Line 56 and throughout - is the instrument called the "MSDs Severity and Frequency Questionnaire?" If so - capitalize each word.

Line 59 - Should "surveyed the MSDs" be "surveyed workers about MSDs?"

Line 59 - MSDs not MSDS

Lines 60-68 and throughout:  It might be helpful to give a brief overview of the industries. Later in the manuscript you occasionally group industries together, but there's not an explanation as to why or how they are similar to one another.

Line 71 - What does "MSDs experiences" mean - clarify phrasing.

Line 107-109: Please specify what "times" meant in this study. For example, are multiple instances of pain spread out throughout a single day considered one time or multiple times? Does an instance have to last for a particular length of time in order to count as a separate time? Do instances have to be separated by a certain amount of time to be considered separate times?

Table 1 and throughout: See above comment about how groups were determined. But I am also wondering why groups are used for some analyses and specific industries are used for others. The majority of the results and discussion seem to be focused on industries, not these groupings. Unless there is an important reason to have these, I would consider presenting all of the data by industry and removing groups. Alternatively, you could include both industry and group data.

Figure 2: Is this supposed to show musculoskeletal discomfort or disorders?  (The instrument seems to be for discomfort.) Is the key supposed to show industry group instead of severity? If it is showing severity, how is this figure different from Figure 1, right side?

Line 195: Recommend rephrasing to "...were the most frequently reported."

Line 197: Recommend rephrasing to "...are also the knees/calves..."

Line 217: Should be group B

Lines 259-260: Please elaborate on your limitations, in particular selection bias - especially the impacts of self-selecting into the study. 

Reviewer 2 Report

1.      This paper must be edited for English grammar.  Just within the abstract there are comma splices, incorrect sentence structure, and awkward wording.

2.      Edit paper for verb tense, missing words in sentences.  “Trial” is not a verb (line 60). Use Northeast or NE Thailand consistently.

3.      The introduction confuses MSDs with risk factors (e.g., repetitive posture (line 44)) with MSDs themselves.  Please clarify this.  Overall, the introduction does not build a strong case for this paper.

4.      In Methods, clarify the age for eligibility is it > than 18 years or =/> 18 years?

5.      In Results, editing is necessary to clarify confusion in the first sentence, lines 120-131.  What is meant by Groups A, B, and C (lines 137-9)?

6.      The instrument used was designed for use in Thailand.  An English translation of the instrument is needed.  It appears it asks how many times a day a pain is felt.  However, much of MSD pain is not single sharp pains but aches and soreness.  How is this accommodated in the measurement of pain?

7.      The first 3 figures are confusing.  Figure 1 says it reports frequency and severity.  How is what is reported there for severity different from Figure 2?    Figure 3 has no key to tell what the colors signify.

8.      What is the logic for combining workers into Groups A, B, C (Lines 181-183).  For example, rice and tobacco cultivation are quite different, so one would expect different physical stressors.

9.      Lines 186-187.  Please indicate which group carries rice bags in Table 3 (the issue seems to be that the table is split.  There is no reason for it not to be a single table.)  Not clear what 1,2,3 signifies in table 3. 

10.   With the large sample size and variation across gender, age group, and crop, the authors need to undertake more sophisticated analyses to focus in on the group that experiences particular kinds of pain.  The simple logistic regression does not tell us very much.

11.   One weakness is that the authors did not query workers about the stressors (e.g., prolonged sitting, kneeling, lifting heavy weights above shoulders).  They are simply inferring this from the crop in which they work.  But they give us little info about the organization of work.  Do all laborers do the same things, or is there variety among workers?

12.   On line 235 the authors report on a percentage who say they have no option to stop their work due to pain.  Where do those data come from?  They were not reported in results and the collection of such data was not reported in methods.

13.   In limitations, the authors should include that all MSD data were self reported and there were no physical exams conducted.

14.   Overall, the paper needs more sophisticated data analyses.  It is not clear what it adds to the literature.  The authors should look beyond Thailand to see what researchers in other areas have done to link work tasks to MSDs.

Reviewer 3 Report

General Comments

This is an interesting cross-sectional study assessing the prevalence (types, frequency and severity) of work-related musculoskeletal disorders (MSDs) among 889 agricultural workers (61.6% female) of various agricultural activities (rice, tobacco, sugarcane, rubber, cassava, fruit, vegetable and corn plantation) in Upper Northeastern Thailand. Most of the participants (68%) worked in the rice plantation. A structured questionnaire was used to collect the data, whereas the first part of the questionnaire was the demographic characteristics and type of tasks performed in agricultural activity, and the second part of the questionnaire was the musculoskeletal disorders (MSDs) questionnaire. The interviews were conducted with the individual participant to complete the questionnaire.

This study indicated clearly that WMSDs are highly prevalent in agricultural workers. About 70% of the participants reported MSDs in at least one part of their body. The majority of the agricultural workers (61%) reported experiencing a mild MSDs level. The main areas of the body that experience MSDs, with regards to their severity and pain level, were the knees/calves, lower back, and shoulder. Farmers working onn cassava, vegetable, and sugarcane plantations, reported experiencing the most severe MSDs in the knees/calves. The rice plantation workers reported the largest number of MSD complainant. Workers on rubber and sugarcane fields were more likely to feel knees/calves pain and lower limb pain than those on rice and tobacco plantations. Individuals working in cassava, fruit, vegetable, and corn plantations were more likely to report knees/calves pain and lower limb pain than those working in rice and tobacco plantations.

The manuscript is generally well written. However, the design of this paper should be improved before publishing. In my opinion, it is obligatory:

(1)  To to present in Abstract (1) information about the age range and sex of the participants

(2) To present age range of participants in Materials and methods.

(3) To present more clearly exclusion criteria of the study in Materials and Methods because in many previous studies of the prevalence WRMSDs in workers with the following history of psychological problem, diabetes, and overweight i.e. body mass index over 30 were excluded.

(4) To present more limiting factors of this study at the end of Discussion (age- and gender-matched differences of the participants, work experience in agriculture, and physical activity of the participants wws not evaluated).

Specific Comments

Abstract

Please add the information about the age range and sex of the participants.  

2. Materials and methods

Please add the information about: (1) the age range of the participants; (2) the exclusion criteria of this study (see General Comments).

3. Results

Figure 3, page 6. Please explain which body parts are presented by different colours in this Figure.

4. Discussion

Please present more limiting factors of this study at the end of Discussion (see General Comments)

Round 2

Reviewer 1 Report

Overall, good improvement.

I still feel the description of "times" (lines 116-118) are a bit unclear - does this now mean that the pain must last for a full day to be considered an instance?

Author Response

Yes, "times" is counted one time of pain. The frequency of pain was considered with reference to the most frequent experience of pain, aches or discomfort in one week or pain which occurred several times throughout a single day. However, if they are multiple instances of pain spread out throughout a single day, they were identified to the highest frequency (several times every day/ persisting for ≥24 hours). You can see more details in an attachment (Appendix A: MSFQ).

Response point (yellow highlight): on line 118-22

Author Response

  1. While the authors have had the manuscript reviewed for English language, they have not hadtheir response to reviewers edited. The manuscript is currently quite readable. However, the response to reviewers is very difficult to comprehend (e.g., response to item 7). So the authors may think they are answering the reviewer queries, but they are not.

Author to respond reviewer: We try to communicate with reviewer and response to reviewers by yellow highlight in the manuscript.

----------------------------------------------------------------------------- 

  1. Make sure that everything in the paper is presented as “frequency” of MSDs first, and “severity”, second. For example, edit the abstract so that is consistent. Also edit the methods to put frequency first.

Author to respond reviewer: We presented as “MSDs severity” first and “MSDs frequency”, second which followed by the sequence of MSFQ.

Response point (yellow highlight): on line 16-7, 51, 57, Figure 1 on line 187

----------------------------------------------------------------------------- 

  1. Throughout, including key words, change “agriculturist” to “agriculturalist”

Author to respond reviewer: Corrective done

Response point (yellow highlight): on line 16-7, 30, 123, 242, 273-4, 281-2, 326-7, 340

----------------------------------------------------------------------------- 

  1. Throughout, “northeast” should be capitalized when it is a noun. When it is an adjective, it should be “northeastern” and not capitalized.

Author to respond reviewer: Corrective done

Response point (yellow highlight): on line 38, 54, 68, 79

----------------------------------------------------------------------------- 

  1. Place the MSFQ translation in an appendix.

Author to respond reviewer: We mention the word “Appendix A” on line 108-10, 261-2, and add the file of Appendix A (attachment file).

----------------------------------------------------------------------------- 

  1. The original critique noted that there did not seem to be a source for data that said the workers  had no option to stop their work due to pain. Although the authors say this is clarified in lines 107-9, it is not. The authors either need to clarify this or they need to include the questionnaire. It is best to give the reader the exact wording of the questions

 Author to respond reviewer: We add the final section of the  questionnaire about pain management in methods, 2.3 Structure questionnaire on line 131-5

Response point (yellow highlight): on line 131-5,The final section of the questionnaire consisted of four questions about how the participants were affected by any aches, pains, or discomfort: (1) Were you able to perform your daily responsibilities? (2) Did you stop working when you experienced any aches, pains, or discomfort? (3) Did you get a Thai massage to release any aches, pains, or discomfort? (4) Did you take any medicine to relieve any aches, pains, or discomfort?”

Reviewer 3 Report

The design of this manuscript was improved in the process of review.

Author Response

Comments and Suggestions for Authors: The design of this manuscript was improved in the process of review.

Author to respond reviewer: Thank you for your comments and suggestions.

Round 3

Reviewer 2 Report

The authors have addressed all the concerns raised in the previous reviews.